# Use of Bio-Waste of *Ilex paraguariensis* A. St. Hil. (Yerba mate) to Obtain an Extract Rich in Phenolic Compounds with Preservative Potential

**DOI:** 10.3390/foods12173241

**Published:** 2023-08-28

**Authors:** Bárbara Menezes, Cristina Caleja, Ricardo C. Calhelha, José Pinela, Maria Inês Dias, Dejan Stojković, Marina Soković, Odinei Hess Gonçalves, Fernanda Vitória Leimann, Eliana Pereira, Lillian Barros

**Affiliations:** 1Centro de Investigação de Montanha (CIMO), Instituto Politécnico de Bragança, Campus de Santa Apolónia, 5300-253 Bragança, Portugal; barbara.menezes@ipb.pt (B.M.); ccaleja@ipb.pt (C.C.); calhelha@ipb.pt (R.C.C.); jpinela@ipb.pt (J.P.); maria.ines@ipb.pt (M.I.D.); lillian@ipb.pt (L.B.); 2Laboratório Associado para a Sustentabilidade e Tecnologia em Regiões de Montanha (SusTEC), Instituto Politécnico de Bragança, Campus de Santa Apolónia, 5300-253 Bragança, Portugal; 3Institute for Biological Research “Siniša Stanković”—National Institute of Republic of Serbia, University of Belgrade, 11000 Belgrade, Serbia; dejanbio@ibiss.bg.ac.rs (D.S.); mris@ibiss.bg.ac.rs (M.S.); 4Department of Post-Graduation Program of Food Technology (PPGTA), Federal University of Technology—Paraná—UTFPR, Campus Campo Mourão, via Rosalina Maria dos Santos, 1233, Campo Mourão CEP 87301-899, PR, Brazil; odinei@utfpr.edu.br (O.H.G.); fernandaleimann@utfpr.edu.br (F.V.L.)

**Keywords:** yerba mate, *Ilex paraguariensis* A. St. Hil., bio-waste, bioactivity, natural ingredients

## Abstract

In this work, a comparison between the extracts of dehydrated yerba mate (*Ilex paraguariensis*) and bio-waste of yerba mate leaves from the Brazilian industry was made. The incorporation of the functional extract as a preservative/functional ingredient in a pastry product (pancakes) was tested. The individual profile of phenolic compounds was determined by HPLC-DAD-ESI/MS, and the bioactive potential was assessed using in vitro assays for antioxidant, anti-inflammatory, antimicrobial, and cytotoxic activities. The yerba mate extracts revealed a high antimicrobial potential against the tested strains and a very promising antioxidant and anti-inflammatory action. Additionally, revealed a cytotoxic capacity for MCF-7, CaCo and AGS tumor cell-lines. In the three types of pancakes, after 3 days of storage, the chemical and nutritional characteristics remain unchanged, proving the preservative efficiency of the extract. This study showed the benefits of the use bio-waste from agro-industrial sector, focusing on sustainable production and the development of circular economy.

## 1. Introduction

Approximately one-third of the food produced annually for human consumption is wasted, amounting to 1.3 billion tons. This causes a growing concern in the agri-food industry regarding the use of discarded waste, which has a high potential for reuse [1].

Several residues resulting from the processing of natural matrices have been identified as excellent sources of phenolic compounds with notable bioactive properties. This biological activity has a special interest in the industrial sector due to its hypothetic use as a functional natural ingredient [2]. A significant part of the residues from the agri-food sector are plants rich in beneficial compounds to health, profitable for the formulation of new products, and, consequently, for the increment of the regional economy. The use of these residues becomes a great alternative for obtaining promising sources of bioactive compounds [3].

Bioactive ingredients, commonly referred to as functional ingredients, are compounds extracted from various natural sources such as plants, fruits, cereals, vegetables, algae, and others. These ingredients retain their characteristics even after the extraction process and incorporation into food products [4]. Thus, to improve the preservative capacity of food and give it a bioactive action, the development of functionalized foods with natural ingredients has instigated a great interest in the scientific community. Consequently, some studies have been developed with the aim of applying natural extracts to food products [5].

The *Ilex paraguariensis* A. St. Hil. (Aquifoliaceae family), commonly known as yerba mate, is a tree with high economic value and significant cultural importance. The leaves are widely consumed in South America through beverage preparations such as tea, “chimarrão” and “tererê” [6]

Considering the amount of solid waste of this plant, during the harvest of raw material, about 5 tons per hectare of branches are discarded in the soil. Additionally, during the grind of the yerba mate, the stalks with greater granulometry are not added to the final product. However, with the purpose of recovery, the smaller stalks are ground again and added to the final product. In this second grinding stage, low-grained residues are generated, called matte powder, which are not added to the final product and are eventually discarded. Anyway, there are many companies that choose to use raw materials with increasingly finer branches and, in most cases, only use the leaves, which results in even greater amounts of plant parts that are not explored/used by the herbal industries [7].

Vieira et al. [8] conducted an analysis of the compounds present in yerba mate processing residues, which proved to be an interesting source of methylxanthines, tannins, and phenolic compounds. Furthermore, they concluded that these residues exhibit excellent antioxidant potential. Additionally, recent research [9] investigated the phenolic compounds present in the extract of *Ilex paraguariensis* and observed the presence of caffeine derivatives (caffeic acid, chlorogenic acid, 3,4-dicaffeoylquinic acid, 3,5-dicaffeoylquinic acid, and 4,5-dicaffeoylquinic acid) and flavonoids (quercetin, rutin, and kaempferol). Authors claimed that this reinforces its potential as a choleretic, antioxidant, and hypocholesterolemic agent.

Yerba mate contains a wide variety of bioactive compounds. However, despite significant knowledge about its composition and health benefits, the market for yerba mate remains limited [10].

Hence, the scientific community, in partnership with the food industry, has shown a high interest in the study of the use of these residues, being a very interesting and promising alternative to obtaining bioactive molecules and natural functional ingredients [11].

Considering the advantages and benefits of using bio-waste from the food industry, this work aimed to study the use of bio-waste from the yerba mate production industry (EMP) (non-standard granulometry commercial acceptance and plant stems) in Brazil to obtain an extract rich in phenolic compounds, capable of being incorporated as a preservative/functional ingredient in a widely consumed and appreciated food product. As a comparative sample, freeze-dried leaves of yerba mate used commercially (EMNP) were also studied. Research works on yerba mate bio-wastes and their application as natural ingredients in pastry products are scarce, differentiating this work from other studies described in the literature.

## 2. Materials and Methods

### 2.1. Plant Material

For this study, two samples of *Ilex paraguariensis* A. St. Hil. of different typologies were used. The sampling is composed of bio-waste of *I. paraguariensis* (EMP) (samples of yerba mate with granulometry that differ from commercial standards and plant stems) and lyophilized leaves of *I. paraguariensis* (EMNP) (Freeze dryer Lioytop L101, Liobrás), which served as a comparison.

The samples were kindly provided in November 2019 by “Terra Mate Indústria e Comércio Ltd., Cascavel, Brazil” a company located in Cascavel, Paraná, Brazil. The samples were ground to a fine powder and then stored in a cool and dry place, protected from light, for later laboratory analysis.

### 2.2. Determination of the Individual Phenolic Profile of Bio-Residues and Lyophilized Leaves of Yerba mate

#### 2.2.1. Extraction Procedure

Two extracts were prepared from the lyophilized samples (one of them from bio-residues of *I. paraguariensis* and the other from dry leaves of *I. paraguariensis*; 1 g). A maceration extraction with ethanol/water solution (80:20, *v*/*v*; 30 mL) at room temperature was applied [12]. The alcoholic fraction of the extracts were evaporated under reduced pressure (Büchi R-210, Flawil, Switzerland), and the aqueous fraction was lyophilized (40 °C; FreeZone 4.5, Labconco, Kansas City, MO, USA) for further analysis. A quantity of the obtained dry extract was subsequently re-dissolved in an ethanol/water solution (5 mg/mL) for further chromato-graphic analysis [12].

#### 2.2.2. Analytical Method

All the chromatographic information was obtained using a Dionex Ultimate 3000 UPLC (Thermo Scientific, San Jose, CA, USA), coupled to a diode array detector (280, 330, and 370 nm) and an electrospray ionization mass detector (Linear Ion Trap LTQ XL, Thermo Finnigan, San Jose, CA, USA), working in the negative mode. The chromatographic separation was performed using a Waters Spherisorb S3 ODS-2 C18 (4.6 × 150 mm, 3 μm, Waters, Milford, MA, USA) column at 35 °C. The compounds were identified considering the retention time, UV-Vis, and mass spectra in comparison with available standards. For quantitative analysis, calibration curves were obtained using injecting standard solutions with known concentrations (2.5–100 μg/mL): chlorogenic acid (*y* = 168823x − 161172), quercetin-3-*O*-rutinoside (*y* = 13343x + 76751) and *p*-coumaric (*y* = 301950x + 6966.7), based on UV-Vis signals and using the maximum absorption wavelength of each standard compound. In the case of unavailable commercial standards, the compounds were quantified via a calibration curve of the most similar standard available. The results were expressed as mg/g of extract [12].

### 2.3. Evaluation of Bioactive Properties of Extracts Obtained from Bio-Residues and Lyophilized Leaves of Yerba mate

The lyophilized extract, provided by the extraction procedures (Section 2.2.1), was re-dissolved: (i) in culture medium (10 mg/mL) for antimicrobial activity assay (Section 2.3.1); (ii) in distilled water at a concentration of 8 mg/mL for the evaluation of cytotoxic activity in tumor and non-tumor cell lines and anti-inflammatory activity (Section 2.3.2); and (iii) in a hydroethanolic solution (ethanol/water; 80:20, *v*/*v*), in a concentration of 5 mg/mL, for antioxidant activity evaluation (Section 2.3.3). These solutions were diluted successively to obtain the working concentrations.

#### 2.3.1. Antimicrobial Activity

The antibacterial activity was evaluated using several bacterial strains according to their importance in terms of food. Gram-negative (*Escherichia coli* (ATCC—American type culture collection 25922), *Salmonella enterica subsp. enterica* serovar Typhimurium (ATCC 13311) and *Enterobacter cloacae* (ATCC 35030)) and Gram-positive bacteria strains (*Staphylococcus aureus* (ATCC 11632), *Bacillus cereus* (clinical isolate), *Listeria monocytogenes* (NCTC—National collection of type cultures 7973). The minimum inhibitory (MIC) and minimum bactericidal (MBC) concentrations were used to estimate antimicrobial potential; the microdilution method was applied. The ampicillin was used as a positive control, and the results were expressed in mg/mL [13].

For the antifungal activity, the fungi tested followed the same selection criteria. *Aspergillus ochraceus* (ATCC 12066), *Aspergillus niger* (ATCC 6275), *Aspergillus versicolor* (ATCC 11730), *Penicillium funiculosum* (ATCC 36839), *Penicillium verrucosum* var. cyclopium (food isolate) and *Penicillium ochrochloron* (ATCC 9112) strains were used. Minimum inhibitory concentration (MIC) and minimum fungicidal concentration (MFC) were determined using a modified microdilution method. Ketoconazole was used as a positive control, and the results were also expressed in mg/mL [13].

#### 2.3.2. Cytotoxic Activity in Tumor and Non-Tumor Cell Lines and Anti-Inflammatory Activity

In order to evaluate the cytotoxicity of the extracts, the sulforhodamine B assay was applied using a methodology previously described by Barros et al. [14]. Five human tumor cell lines were used: MCF-7 (breast adenocarcinoma), NCI-H460 (lung carcinoma), HeLa (cervical carcinoma), AGS (gastric adenocarcinoma), and CaCo-2 (colorectal adenocarcinoma). Monkey kidney non-tumor cell lines (VERO) were also used in order to test the extracts’ toxicity. All human or animal cell lines used in this work are commercially available and were purchased from different authorized cell line resources including: the German Collection of Microorganisms and Cell Cultures (DSMZ) and the European Collection of Authenticated Cell Cultures (ECCAC). These cell lines were obtained from DSMZ, except the CaCo2 cell line that were obtained from the ECACC. In order to maintain high scientific standards, all procedures will be performed according to the best practices observed in the Guidance on Good Cell Culture Practice (GCCP). The results were expressed as GI_50_ values (the extract concentration that inhibits 50% of the cell growth).

For the evaluation of anti-inflammatory activity, RAW 264.7 macrophage cells were employed, as described in the study by Jabeur et al. [12]. The RAW 264.7 cells were purchased from the European Collection of Animal Cell Culture (ECACC) located in Salisbury, UK. Cell cultures were maintained in DMEM, supplemented with 10% heat-inactivated bovine serum and L-glutamine, at 37 °C with 5% CO_2_ in humidified air. The nitric oxide produced was determined by measuring the absorbance at 540 nm (ELX800 BioTek microplate reader) and compared to the calibration curve.

#### 2.3.3. Antioxidant Activity

For the antioxidant activity evaluation, two assays were used, using methodologies previously described [15,16]: the cell-based assays of thiobarbituric acid reactive substances (TBARS) formation inhibition and oxidative hemolysis inhibition (OxHLIA). The extracts’ capacity to inhibit the formation of TBARS was assessed using porcine brain cells as oxidizable biological substrates, and the results were expressed as EC_50_ values (g/mL), which correspond to the concentration of the extract responsible for 50% of antioxidant activity. In turn, the extracts’ capacity to inhibit the oxidative hemolysis was tested using sheep blood erythrocytes as ex vivo models, and the extract concentration able to promote a Dt hemolysis delay of 60 min was calculated based on the Ht_50_ values of the hemolytic curves of each extract concentration. The results were expressed as IC_50_ values (µg/mL), which represent the extract concentration required to keep 50% of the erythrocyte population intact for 60 min. Trolox was used as a positive control in both assays.

### 2.4. Incorporation of Phenolic Compound-Rich Extract into Pancakes

For the incorporation of an ingredient with preservative potential, the extract of yerba mate bio-residues (EMP) was chosen due to the higher content in most of the phenolic acids detected and the interest in the reuse of industrial bio-wastes.

#### 2.4.1. Preparation of the Pastry Product—Pancakes

The incorporation was made in pancakes, a widely appreciated and consumed food product. To prepare the pancakes, a traditional recipe was followed: two eggs were thoroughly mixed with 17 g of sugar, 11 g of baking powder, and 210 mL of milk. Then, 184 g flour was sequentially added to the mixture while mixing vigorously with a hand mixer at 450 W for 8 min (Bosch, Munich, Germany). Three samplings of pancakes were prepared, one of them without adding any type of additive, another with the addition of the preservative functional ingredient, and another with potassium sorbate (the preservative added by the food industry), for commercial purposes. The amount of natural ingredient, as well as the amount of potassium sorbate, was selected considering the maximum permitted dose legally legislated—25 mg/kg [17].

After incorporation, the preservative power of the selected extract (ingredient rich in phenolic compounds with preservative action) was tested at 0 and 3 days of storage (considering the lifetime of a homemade preparation) in a dry, cool, and dark place and, subsequently, physical, nutritional and chemical evaluations were carried out to test the viability of the added additives. Samples with the addition of extract rich in phenolic compounds (AEO) were compared with control samples (without the addition of any type of additive—AC) and with samples incorporated with a synthetic additive commonly applied in this type of product (potassium sorbate—ASP).

#### 2.4.2. Evaluation of the Color Parameters of Pancake Samples during Storage Time—0 and 3 Days

The color measurement was performed to assess possible differences after incorporation. A colorimeter (model CR-400; Konica Minolta Sensing, Inc., Tokyo, Japan) coupled to an adapter for granular materials (model CRA50) was used, following the methodology previously described by Roriz et al. [18].

The values of the three-dimensional coordinates CIE L* a* b* were obtained in a computerized system with a type C illuminant and an 8 mm diameter diaphragm, and for data processing, the Spectra Magic Nx software (CM-S100W 2.03.0006 version, Konica Minolta, Japan) was used. Regarding the three-dimensional coordinates obtained, L* represents luminosity, a* represents chromaticity on an axis from green (−) to red (+), and b* represents chromaticity on an axis from blue (−) to yellow (+).

#### 2.4.3. Evaluation of the Nutritional and Chemical Composition of Pancake Samples at 0 and 3 Days of Storage

The protein, fat, carbohydrates, and ash content of the pancake samples were obtained considering AOAC [19] procedures and using methodologies described by Barros et al. [14]. For the crude protein (N × 5.70), a Kjeldahl method (AOAC 978.04) was applied, and the ash content was obtained by exposing the sample to incineration at 550 ± 15 °C for 12 h (AOAC 923.03), whereas the crude fat was obtained using a Soxhlet apparatus with petroleum ether as recycling solvent (AOAC 920.85) and, finally, the total carbohydrate was assessed using differences. To determine the total energy, the following equation was used: Energy (kcal) = 4 × (g protein + g carbohydrates) + 9 × (g fat).

The chemical composition of the pancake samples was determined by analysis of sugars and fatty acids, according to methodologies previously described by Barros et al. [14], and analyzed using chromatographic systems, namely, HPLC-RI and GC-FID, respectively. The compounds were characterized by comparison with available standards (standard 47885, Sigma-Aldrich, St. Louis, MO, USA). The content in sugars was expressed in g/100 g of fresh weight, and melezitose (Sigma Chemical Co.; Saint Louis, MO, USA) was used as an internal standard in sugars evaluation. The fatty acids concentration was expressed as relative percentages (%) of each fatty acid.

### 2.5. Statistical Analysis

The tests mentioned in this study were performed in triplicate, and the results were expressed as mean ± standard deviation (SD). The statistical analysis of the data was performed in order to determine the significant differences between the different samples and was carried out using an analysis of variances: one-way ANOVA and the *t*-student test, considering the different types of comparisons. In each table, the statistical test applied was described. For this, the SPSS v program was used. 23.0 (IBM Corp., Armonk, New York, NY, USA).

## 3. Results

### 3.1. Individual Phenolic Profile of Samples from Ilex paraguariensis A. St. Hil.

The detailed individual profile of the phenolic compounds present in the lyophilized leaves (EMNP) and in the bio-residues of yerba mate (EMP) is shown in Table 1.

The identification of the compounds was based on retention times (Tr), maximum absorption wavelengths in the region of UV-Vis (ʎ_max_), pseudomolecular ion ([M-H]^−^), and fragmentation of the molecular ion (MS^2^), comparing with the standards available in the literature. Seven phenolic compounds were identified, including six phenolic acids: 4-*O*-caffeoylquinic acid, 5-*O*-caffeoylquinic acid (chlorogenic acid), 4-*O*-*p*-coumaroylquinic acid, 3,4-*O*-dicaffeoylquinic, 3,5-*O*-dicaffeoylquinic acid, and 4,5-*O*-dicaffeoylquinic acid; and one flavonoid: quercetin-3-*O*-rutinoside.

The total content of phenolic compounds ranged between 9.31 ± 0.19 mg/g extract for the EMP sample and 6.75 ± 0.24 mg/g extract for the EMNP sample. In the EMP sample, 2.60 ± 0.16 and 2.12 ± 0.03 (peaks 2 and 6) stood out as the majority compound (5-*O*-caffeoylquinic acid and 3,5-*O*-dicaffeoylquinic acid, respectively); contrary to the EMNP sample, 3,5-*O*-dicaffeoylquinic acid and 4-*O*-*p*-coumaroylquinic acid (peaks 6 and 3) stood out, with values of 1.97 ± 0.14 and 1.92 ± 0.04 mg/g extract, respectively.

The total concentration obtained for phenolic acids ranged between 8.99 ± 0.18 and 6.09 ± 0.23 mg/g extract for EMP and EMNP, respectively.

Considering the statistical analysis, significant differences (*p*-value < 0.05) were evident for most of the identified compounds. The variation of concentrations depended on the evaluated sample: EMP (*I. paraguariensis* bio-residues) or EMNP (lyophilized *I. paraguariensis* leaves). A statistically significant difference (*p*-value < 0.05) was observed between samples in 4-*O*-caffeoylquinic acid (peak 1), 5-*O*-caffeoylquinic acid (peak 2), quercetin-3-*O*-rutinoside (peak 4), 3,4-*O*-dicaffeoylquinic acid (peak 5), 3,5-*O*-dicaffeoylquinic acid (peak 6), 4,5-*O*-dicaffeoylquinic acid (peak 7), as well as the total flavonoid content. On the other hand, no statistically significant differences (*p*-value > 0.05) were observed in 4-*O*-*p*-coumaroylquinic acid (peak 3), the total phenolic content, and the total content of phenolic acids, with *p*-values of 0.239, 0.599, and 0.492, respectively.

The EMP sample exhibited higher concentrations of most detected phenolic acids, such as 4-*O*-caffeoylquinic acid, 5-*O*-caffeoylquinic acid, 3,4-*O*-dicaffeoylquinic acid, 3,5-*O*-dicaffeoylquinic acid, and 4,5-*O*-dicaffeoylquinic acid, compared to the EMNP sample. Conversely, the EMNP sample displayed a higher total flavonoid content. 

### 3.2. Bioactive Properties of Extracts Obtained from Bio-Residues and Lyophilized Yerba mate Leaves

#### 3.2.1. Antimicrobial Activity

The spectrum of antimicrobial action of natural products is broad, comprising Gram-positive and Gram-negative microorganisms [13]. The results obtained from these assays, which encompassed antibacterial and antifungal assessments of hydroethanolic extracts from lyophilized leaves (EMNP) and bio-residues (EMP) of *I. paraguariensis*, are presented in Table 2.

The EMP and EMNP extracts showed antimicrobial activity against the studied strains. The minimum inhibitory concentration (MIC) for *Staphylococcus aureus* was 1 mg/mL, while the minimum bactericidal concentration (MBC) was 2 mg/mL for both extracts. For *Bacillus cereus*, the MIC was lower for the EMP extract (0.5 mg/mL) compared to the EMNP extract (1 mg/mL), but the MBC values were the same (2 mg/mL) for both. *Listeria monocytogenes* were more susceptible to the EMNP extract (MIC = 1 mg/mL) than to the EMP extract (MIC = 2 mg/mL), while the MBC values were equal (2 mg/mL) for both extracts. Both extracts exhibited high bactericidal potential against *Escherichia coli*, with MBC values of 0.5 mg/mL. The *Salmonella* Typhimurium strain showed MIC and MBC of 2 mg/mL for the EMP extract and MIC of 1 mg/mL, and MBC of 2 mg/mL for the EMNP extract. For *Enterobacter cloacae*, the results were less promising for both extracts, with MIC of 1 mg/mL and MBC of 2 mg/mL for the EMNP extract, and MIC of 2 mg/mL and MBC of 4 mg/mL for the EMP extract.

Regarding the antifungal potential, it is evident that the MIC and MFC values of the strains *Aspergillus ochraceus*, *Aspergillus niger*, *Penicillium funiculosum,* and *Penicillium verrucosum* var. cyclopium strains were the same for the EMP and EMNP extracts, with MIC values equal to 0.5 mg/mL and MFC equal to 1 mg/mL. The most promising antifungal activity occurred against the *Aspergillus versicolor* and *Penicillium ochrochloron* strains, where the species most susceptible to extracts was *Penicillium ochrochloron*, with MIC and MFC values (0.25 mg/mL) in the EMP extract and MIC and MFC (0.125 mg/mL) for the EMNP extract. The *Aspergillus versicolor* was also susceptible to EMP and EMNP extracts, with MIC (0.25 mg/mL) and MFC (0.5 mg/mL).

#### 3.2.2. Anti-Proliferative Activity in Tumor and Non-Tumor Cell and Anti-Inflammatory Activity

Toxicity is one of the crucial parameters for the evaluation of biological response and the harmful potential that can lead to the death of cells and tissues [20]. Tumor and non-tumor cell lines were employed to assess the anti-proliferative activity of the extracts. The lyophilized extracts obtained were separately dissolved in distilled water at a concentration of 8 mg/mL for evaluating cytotoxic and anti-inflammatory activities. These results are presented in Table 3. 

Considering the results in the tumor cell lines, the anti-proliferative capacity of both extracts was evident only in the MCF-7, CaCo-2, and AGS cell lines; for HeLa and NCI-H460, the EMP and EMNP extract showed no activity (GI_50_ > 400 µg/mL). 

For the MCF-7 tumor cell line, GI_50_ values were found to be approximately 196 µg/mL for EMP and 159.5 µg/mL for EMNP. In the CaCo-2 tumor cell line, GI_50_ values were approximately 162 µg/mL for EMP and 85.3 µg/mL for EMNP. Additionally, in the AGS tumor cell line, GI_50_ values were close to 201 µg/mL for EMP and 238 µg/mL for EMNP. 

In the non-tumor cell line that served to test the toxicity of the extract, that is, the safety of its incorporation, was monkey kidney cells (VERO). The results indicated that the EMP and EMNP extracts did not exhibit toxicity at the maximum concentration tested (GI_50_ > 400 μg/mL). Regarding the anti-inflammatory activity, both extracts showed a promising action, with values of 302 ± 10 μg/mL for the EMP extracts and 258 ± 8 μg/mL for EMNP extracts. 

#### 3.2.3. Antioxidant Activity

To evaluate the antioxidant activity of the hydroethanolic extract of the lyophilized leaves and the bio-residues of *I. paraguariensis*, two in vitro methods were applied (lipid peroxidation inhibition—TBARS and oxidative hemolysis inhibition—OxHLIA), and the results are shown in Table 4.

Considering the EC_50_ values obtained in TBARS, both extracts revealed antioxidant potential. EMNP proved to be the extract with the best antioxidant activity (5.8 ± 0.2 µg/mL). Otherwise, in the OxHLIA tests in 60 and 120 min, the EMP extract demonstrated greater antioxidant capacity.

### 3.3. Study of Incorporation of Phenolic Compound-Rich Extract into Pancakes 

#### 3.3.1. Evaluation of Color Parameters of Pancakes

This study aimed to assess the impact of incorporating an extract rich in preservative compounds on the external appearance of pancakes. Specifically, the color parameters, including L* (luminosity), a* (red color intensity), and b* (yellow color intensity), were evaluated in three sample groups: the control sample, the sample enriched with the extract rich in preservative compounds, and the sample with the addition of the potassium sorbate (synthetic preservative). The comparisons were made based on the method of incorporation. A particular focus was placed on comparing the control sample with the sample enriched with the extract after a 3-day storage time. The results are shown in Table 5.

After the statistical analysis of the data, a significant variation was evident in some of the evaluated parameters. Considering day 0, a significant variance was observed in the L* and a* parameters (*p* < 0.05), with values ranging from 69.0 ± 2.3 to 76.7 ± 1.1 and from 2.4 ± 0.1 to 4.9 ± 0.2, respectively. Regarding the samples after 3 days of storage, the same variation in the parameters was also observed, with only the b* parameter showing no statistically significant changes and ranging from 28.1 ± 1.1 to 28.8 ± 1.4. 

#### 3.3.2. Evaluation of the Nutritional Value and Chemical Composition of Pancakes

Table 6 presents the nutritional composition and energy content, expressed in grams per 100 g of fresh weight (fw), for the three types of pancakes: control pancakes (AC), pancakes enriched with the extract rich in phenolic compounds (AEO), and pancakes with the addition of potassium sorbate (ASP). Additionally, the evaluation of free sugars and fatty acids is expressed in the same table.

The samples with the addition of extract (AEO), at t = 0 days, showed very similar humidity contents (51.9 ± 0.1 g/100 g fw) to the control samples (AC) (51.1 ± 0.1 g /100 g fw) and from the samples with the added potassium sorbate (ASP) (50.7 ± 0.1 g/100 g fw). In terms of protein content, the ASP samples (7.2 ± 0.1 g/100 g fw) had higher concentrations than the AEO samples (7.14 ± 0.01 g/100 g fw) and AC (7.1 ± 0.2 g/100 g fw) at t = 0 days.

Regarding ash content, the AC samples and the AEO samples had oscillations between 2.5 ± 0.1 g/100 g fw of ash at t = 0 days, while the ASP samples showed contents 2.490 ± 0.003 g/100 g fw. The fat content had little variation among the samples, with values very close at t = 0 (AC = 3.17 ± 0.4 g/100 g fw ± 0.04; AEO = 3.2 ± 0.1 g/100 g fw; ASP = 3.2 ± 0.2 g/100 g fw).

Carbohydrates were the most abundant nutrients in all samples. At t = 0 days, the carbohydrate fluctuations in AC, AEO, and ASP samples were 36.1 ± 0.2 g/100 g fw, 35.35 ± 0.03 g/100 g fw, and 36.4 ± 0. 1 g/100 g fw, respectively.

The ASP samples showed the highest energy content at t = 0 (203.1 ± 1.1 Kcal/100 g), while the AEO samples were less energetic (198.4 ± 1.1 Kcal/100 g). After 3 days, the AEO samples had the highest energy content (201.3 ± 0.2 Kcal/100 g), and the AC samples had the lowest (194.1 ± 0.8 Kcal/100 g).

Considering the free sugars profile, the composition was also evaluated in all samples (AC, AEO, and ASP). It was observed that only one disaccharide (sucrose) was identified, representing the total sugar content. The sugar values ranged from 21.3 ± 0.9 to 25.0 ± 0.5 mg/100 g fresh weight at time 0 days and from 20.1 ± 0.7 to 24.0 ± 0.6 mg/100 g fresh weight at time 3 days.

The results of the lipid fraction analysis of the samples (AC, AEO, and ASP) showed the presence of 20 fatty acids, with only the major fatty acids (with contents above 7%) presented in Table 6. Oleic acid (C18:1n9) was the most abundant fatty acid in all analyzed extracts, followed by palmitic acid (C16:0), linoleic acid (C18:2n6), and stearic acid (C18:0), at both tested storage times (t = 0 days and t = 3 days). Furthermore, the major class of fatty acids in the AC, AEO, and ASP samples was saturated fatty acids (SFA), followed by monounsaturated fatty acids (MUFA) and polyunsaturated fatty acids (PUFA) at both storage times.

## 4. Discussion

### 4.1. Phenolic Profile of Samples from Ilex paraguariensis A. St. Hil.

Phenolic compounds are a class of bioactive compounds present in many plant matrices, particularly in *Ilex paraguariensis,* most associated with antioxidant properties, helping human health [21]. The EMP sample presented the 5-*O*-caffeoylquinic acid (2.60 ± 0.16 mg/g) as the major compound, the main representative of the chlorogenic acid family. This compound is produced in plants from an ester bond between the carboxyl group of caffeic acid and the 5-hydroxyl group of quinic acid [22]. Chlorogenic acid and caffeic acid are linked to decreased risks of chronic diseases, namely inflammation, cardiovascular disease, and cancer. The 5-*O*-caffeoylquinic isomer showed anti-inflammatory properties due to its inhibitory capacity in the inflammatory process mediated by cytokines [23].

The EMNP sample showed 3,5-*O*-dicaffeoylquinic acid (1.97 ± 0.14 mg/g) as the major compound. Dicaffeoylquinic acids are formed by the esterification of hydroxycinnamic acids with quinic acid [24]. Caffeoylquinic acid derivatives, in general, have varied pharmacological activities, with the recent discovery of a neuroprotective effect that reduces oxidative stress associated with degenerative diseases, including Parkinson’s and Alzheimer’s diseases [23].

Other authors also studied the individual profile of phenolic compounds in *I. para-guariensis* samples. Filip et al. [25] used an HPLC-PAD/UV method to identify and quantify the total content of caffeoyl derivatives and flavonoids present in aqueous extracts obtained by decoction of *I. paraguariensis*. As in the present study, these authors also found the greatest abundance of phenolic acids, with 3,5-*O*-dicaffeoylquinic acid being found in higher quantity, followed by 4,5-*O*-dicaffeoylquinic acid and chlorogenic acid, with concentrations of 3.040 ± 0.180 (% by dry weight); 2.890 ± 0.060 (% by dry weight) and 2.800 ± 0.300 (% by dry weight), respectively.

Pagliosa [11] also made a comparative study between the phenolic compounds of the leaves and bark of *I. paraguariensis* branches (bio-residues), applying an aqueous and methanolic extraction. The results showed that both extractions obtained similar chromatographic profiles in relation to the detected phenolic acids. However, the concentrations were higher in yerba mate husks, with emphasis on 4,5-*O*-dicaffeoylquinic acid, with six times more amount (5177.77 ± 603.49 mg/100 g) compared to leaves (896.39 ± 76.89 mg/100 g) and followed by chlorogenic acid with double amount (2928.25 ± 68.01 mg/100 g in the peels and 1608.23 ± 5.85 mg/100 g in the leaves). In the present work, although the extraction was carried out in a hydroethanolic solution (80:20, *v*/*v*), the amount of 4,5-*O*- dicaffeoylquinic acid and chlorogenic acid (5-*O*-caffeoylquinic acid) was also higher in the bio-residues of *I. paraguariensis* when compared to lyophilized leaves.

Cardozo et al. [26] present a study of methylxanthines and phenolic compounds in progenies of yerba mate leaves (*I. paraguariensis*), applying a hydroethanolic extraction (70:30, *v*/*v*) and the detection carried out in an HPLC-UV system. The results revealed that the average contents (%) of phenolic compounds present in the samples are mostly chlorogenic acids (between 0.786 and 0.932%), followed by caffeic acids (between 0.014 and 0.020%). It is necessary to emphasize that chlorogenic acids are metabolic products of phenylpropanoids and are associated with the plant’s response to environmental changes; thus, Cardozo et al. [26] concluded that soil composition can influence both the chloro-genic acid content and the total phenol content of yerba mate.

According to Filip et al. [25], *I. paraguariensis* is known for its high content of hydroxycinnamic acid group compounds, with emphasis on caffeic acid derivatives. Bravo et al. [27] stated that commercial yerba mate has a high content of hydroxycinnamic acid due to the large amount of leaves present. This statement can be confirmed in this research, which also reveals that bio-residues have the same characteristics as lyophilized leaves, in addition to having phenolic acids in major concentration.

Compared to the previous studies with the results obtained in the present study, a disparity in the phenolic profile was identified. This difference can be explained by several factors, such as the possibility of interaction of phenolic compounds with other components of the plant and forming insoluble complexes. The solubility of these compounds is also affected by the polarity of the solvent used, making it difficult to apply an adequate extraction procedure to obtain all phenolic compounds [28]. The extraction yield of soluble compounds from the aqueous extract of yerba mate depends on the water temperature, the degree of subdivision of the herb, and the effective contact of the phases [29].

Other factors can affect the chemical composition of plant matrices, especially genetic variability [10] in the environmental and cultivation codes: geographical origin, precipitation, elevation, soil composition, air pollution, and exposure to the sun [30]. Consequently, post-harvest processes such as sapeco and roasting performed on yerba mate lead to structural changes in its constituents [31]. The chemical composition is also affected by the processing conditions (dry or roasted leaves) and the infusion method used in the preparation of the drink (tea, chimarrão, or tererê) [32].

### 4.2. Bioactive Evaluation of Extracts Obtained from Bio-Residues and Lyophilized Yerba Mate Leaves

#### 4.2.1. Antimicrobial Activity

Considering the antimicrobial potential, the results obtained highlighted the antibacterial capacity against the studied strains. The analysis revealed a high bactericidal potential against *Escherichia coli* strains, with MBC values of 0.5 mg/mL in both extracts.

Several studies have been carried out to evaluate the antimicrobial potential of *Ilex paraguariensis*. The biological activity observed in the hydroalcoholic extracts may be related to the concentration of bioactive compounds (possibly phenolic compounds) existing in the composition of this plant species, which were previously described by other authors as having high antimicrobial activity [23]. One of the potential mechanisms of action of hydroxycinnamic acids is their ability to interact with the cell membrane of microorganisms, inducing damage and alterations to their structure. This membrane disruption may lead to the release of vital cellular components, ultimately resulting in the demise of the microorganism [33].

According to [23], the chlorogenic 5-*O*-caffeoylquinic can exert a beneficial effect on the harmful intestinal microbiota found in the colon, making it a promising option as a food preservative and additive. Additionally, Ref. [23] highlights that the caffeoylquinic acids, especially 5-*O*-caffeoylquinic, have properties that can render them potential natural agents with antibacterial, antifungal, and antiviral activities. These characteristics may open intriguing possibilities for the utilization of these compounds in food preservation and the formulation of food products with added health benefits. For example, Ref. [23] evaluated the antimicrobial activity of hydromethanolic and hydroethanolic extracts of *I. paraguariensis*, obtained by percolation, against food pathogens (*S. aureus*, *L. monocytogenes*, *S. Enteritidis* and *E. coli*), using gas chromatography coupled with mass spectrometry. Both extracts inhibited the growth of the tested strains, except for *E. coli*. For the hydroethanolic extract (40:60, *v*/*v*), the result of MIC and MBC was 3.13 mg/mL and 6.25 mg/mL for *S. enteritidis*, 0.78 mg/mL (MIC and MBC) for *S. aureus* and 3.13 mg/mL (MIC and MBC) for *L. monocytogenes*. These authors had better results with the hydroethanolic extraction and were able to conclude that the phenolic contents of the extracts are directly related to their antimicrobial power.

In a study carried out by [34] regarding the antimicrobial activity of extracts of leaves and branches of *I. paraguariensis*, it was detected inhibitory activity in the hydroethanolic extract of yerba mate against several microorganisms; however, only in the *E. coli* strain was there no inhibition. Additionally, the antimicrobial potential of yerba mate hydroethanolic extract for preservative purposes was also studied by [35], where the objective was to be incorporated into fish hamburgers. The results obtained in the MIC assay were 10 mg/mL for *E. coli* and *S. typhi* and 5 mg/mL for *S. aureus*. With the results, it was possible to verify that the use of the yerba mate extract in the control of microbial growth in fish burgers is a promising proposal for food preservation.

The extraction method applied and the solvent used are factors of high importance for the determination of antimicrobial activity. Alcoholic solvents are capable of breaking cellular structures, such as membranes, and extracting intracellular compounds; however, within the same plant species, the content of constituents of active groups can vary substantially [36]. This explains the different concentrations found in the extracts.

#### 4.2.2. Anti-Proliferative Activity in Tumor and Non-Tumor Cell Lines and Anti-Inflammatory Activity

According to Table 3, it was clear that the studied extracts (80:20, *v*/*v*) obtained from the lyophilized leaves and the bio-residues of *I. paraguariensis* did not show anti-proliferative capacity in the tumor lines HeLa and NCI-H460, presenting values of GI_50_ > 400 µg/mL. Based on the statistical treatment applied, it was evident that in all tumor cell lines where inhibitory activity was observed, there were also statistically significant differences considering the type of sample tested. In this context, lyophilized leaves of *I. paraguariensis* (EMNP), when compared to bio-residues (EMP), exhibited better inhibitory capacity in MCF-7 and CaCo tumor lines. Moreover, in both extracts, the highest inhibitory potential was observed in the AGS line.

Regarding the anti-inflammatory activity, the results suggest the presence of anti-inflammatory potential in both extracts. The applied statistical evaluation also revealed a significant difference between the samples, with particular emphasis on the lyophilized leaves of *I. paraguariensis* (EMNP) that demonstrated superior anti-proliferative capacity for RAW 264.7 cells. Inflammation is a complex physiological process of tissue injury caused by exogenous or endogenous sources. Prolonged dysregulated inflammatory processes can lead to tissue damage and are the underlying cause of many chronic pathologies, such as diabetes, alcoholic liver disease, chronic kidney disease, and cardiovascular and neurodegenerative disorders [37]. The chlorogenic acids, especially 5-*O*-caffeoylquinic, have demonstrated anti-inflammatory activity by reducing pro-inflammatory cytokines through the modulation of key transcription factors such as tumor necrosis factor-alpha (TNF-α) and interleukins, such as IL-8 [23].

A study was carried out by [38] about the cytotoxicity, hepatotoxicity, and anti-inflammatory activity using different parts of *I. paraguariensis* (whole plant, leaves, and stems). Four tumor cell lines were used for cytotoxicity analysis: MCF-7, NCI-H460, HeLa, and HepG2. All extracts showed anti-proliferative capacity in the tumor lines tested; however the extract from the yerba mate stem was more potent against MCF-7 and HepG2 cell lines. The hepatotoxicity of the extracts was evaluated in non-tumor porcine liver lines (PLP2), and the results were expressed as GI_50_ values that were >400 μg/mL, proving the absence of toxicity. These authors also confirmed the anti-inflammatory activity of the obtained extracts, with emphasis on the extract from the stem of *I. paraguariensis,* where they obtained the lowest GI_50_ content (26 ± 1 μg/mL). 

Additionally, Ref. [39] evaluated the effect of yerba mate tea extract in vitro and in vivo models of ethanol-induced liver injury in rats and obtained positive results, since the extract was able to supply the increase in cell death by inhibiting the cytochrome p450 2E1 (CYP2E1) activity, leading to the conclusion that yerba mate tea extract can prevent alcohol-induced liver damage. Additionally, Ref. [40] found the in vitro anti-inflammatory effect of methanolic extracts of *Ilex paraguariensis* against inflammation induced by *Propionibacterium acnes*. Furthermore, Ref. [41] demonstrated that yerba mate extract has potential anti-obesity effects on adipose tissue due to its action in reducing obesity-associated inflammation.

Considering the results obtained in this work, both extracts showed promising results regarding the cytotoxic and anti-inflammatory potential. However, the lyophilized leaf extract had better results, as expected.

#### 4.2.3. Antioxidant Activity

The antioxidant activity of plant extracts is commonly related to the presence of bioactive molecules in their composition. Some of these molecules may be phenolic compounds, which are characterized by being able to donate hydrogen radicals to pair with other available free radicals, which can delay oxidation and stabilize the system [35]. 

Caffeoylquinic acids, such as chlorogenic acid, exhibit antioxidant activity due to the presence of phenolic groups in their structure. These compounds donate electrons to free radicals, neutralizing them and protecting cells against oxidative damage. Additionally, they stimulate endogenous antioxidant enzymes, further enhancing the body’s antioxidant defense. This antioxidant activity may contribute to the prevention of diseases related to oxidative stress and cellular aging [23]. Considering the applied statistical treatment, the significative differences detected in the OxHLIA assay can be explained by the presence of a major concentration of total phenolic acids and total phenolic compounds previously verified in these samples.

Contrary to the results obtained in this study, Ref. [42] evaluated the antioxidant activity of the hydroethanolic extract (50:50, *v*/*v*) of yerba mate using the DPPH assay and obtained an IC_50_ value of 0.37 mg/mL. On the other hand, Ref. [35] studied the antioxidant activity of the hydroethanolic extract of yerba mate, also using the DPPH assay, and reached an IC_50_ value = 7.91 ± 0.06 µg/mL, demonstrating a good antioxidant activity of the extract.

In a study carried out by [38] regarding the phytochemicals and bioactive properties of individual parts of *I. paraguariensis*, for TBARS inhibition, the EC_50_ values obtained for the whole plant, leaves, and stems were 61 ± 1 µg/mL, 60 ± 3 µg/mL, and 290 ± 24 µg/mL, respectively. Compared with the present study, the results for the antioxidant activity of the bio-residue extract of *I. paraguariensis* showed an even greater antioxidant capacity, confirming the efficiency of the extract.

### 4.3. Study of Incorporation of Phenolic Compound-Rich Extract into Pancakes 

#### 4.3.1. Evaluation of Color Parameters of Pancakes

In the incorporation study, after statistical analysis of the data, a significant variation was evident in some of the parameters evaluated. There was a change in the L* parameter, which could be attributed to the cooking time that resulted in increased darkening of both the control sample and the sample with the addition of potassium sorbate. It is important to consider that this was a laboratory test, not an industrial process conducted in a controlled setting. 

Furthermore, when the *t*-student test was applied to verify the variance between the sample with the addition of the preservative extract at 0 and 3 days, no significant differences were observed in any parameter, as the *p*-value values were higher than 0.05. This allows us to verify the absence of color changes in pancakes with the phenolic-rich extract after 3 days of storage (representing the domestic shelf life of this food product), indicating that it is a viable alternative for replacing the commercially applied preservative—potassium sorbate.

The values expressed in total color difference (∆E) between days 0 and 3 showed that the control samples (AC) had greater color variations compared to the AEO and ASP samples, whose values indicate the same variation between the two times. This result assumes that the extract rich in phenolic compounds is as effective as potassium sorbate.

#### 4.3.2. Evaluation of the Nutritional Value and Chemical Composition of Pancakes

After statistical analysis, it was observed that the small variations in the various analyzed parameters are not significant when compared (*p* > 0.05). This indicates that the different types of pancakes have the same nutritional profile. Additionally, after 3 days of storage, the pancake samples with the addition of the natural additive maintained a similar nutritional profile (*p* > 0.05) without undergoing any food product degradation, allowing for its preservation.

In the statistical analysis of the free sugars profile, a significant variation (*p* < 0.05) was observed among the different samples. The pancake variant enriched with the extract rich in phenolic compounds (AEO) showed higher sucrose content, with values of 25.0 ± 0.5 at t = 0 days and 24.0 ± 0.6 at t = 3 days. Through the analysis of the storage time of the AEO sample, the absence of significant differences in the obtained values (*p* > 0.05) shows the maintenance of the free sugar profile, that is, showing the conservation potential of this food product.

Regarding the statistical analysis of the lipid fraction, significant variations were observed in some of the identified fatty acids, as well as in the levels of monounsaturated fatty acids (MUFA) and polyunsaturated fatty acids (PUFA). However, it is important to note that these changes might be attributed to the homemade preparation process, where industrial standards for rigor in cooking and ingredient homogenization could not be fully controlled, leading to potential differences in the chemical and nutritional composition among the pancake samples.

## 5. Conclusions

The extract obtained from the lyophilized leaves of yerba mate had 3,5-*O*-dicaffeoylquinic acid as the major compound, while the extract obtained from the bio-residues had 5-*O*-caffeoylquinic acid as the predominant compound.

Regarding the evaluation of bioactivities, the hydroethanolic extracts showed antimicrobial capacity against the studied strains, having a high bactericidal potential against the strains *Escherichia coli* and high fungicidal potential against the strains of *Penicillium ochrochloron*. The cytotoxicity tests showed that at the maximum concentrations tested in non-tumor cell lines (VERO), the extracts have no toxicity. Anti-proliferative activity has been demonstrated in the tumor cell lines MCF-7, CaCo-2, and AGS. The anti-inflammatory activity test showed promising values that prove the extract’s anti-inflammatory potential. Finally, the test for antioxidant activity showed the efficiency of the extract, which can be explained due to the high content of phenolic acid in the sample.

The bio-residues of *I. paraguariensis* become suitable for application in the industrial sector, as they have phenolic compounds with preservative capacity in the extracts. The results of the incorporation of the extract rich in phenolic preservative compounds in pancakes, regarding the color, showed promising due to the absence of color changes compared to the other extracts (control and with the addition of potassium sorbate).

Considering the nutritional and chemical parameters evaluated, in general, it was found that the small oscillations were not enough to change the nutritional and chemical profile of the pancakes. In addition, the extract’s preservative capacity after 3 days of shelf life was notorious, showing no changes in the nutritional profile and in the chemical parameters evaluated. As such, this study allowed us to conclude that the incorporation of the extract from *I. paraguariensis* bio-residues in pastry products proves its preservative capacity, being able to be incorporated in the composition of other food matrices.

## Figures and Tables

**Table 1 foods-12-03241-t001:** Retention time (Tr), maximum absorption wavelengths in the UV-Vis region (λ_max_), attempt to identify and quantify phenolic compounds of the yerba mate extracts (mean ± SD).

Peak	Tr (min)	λ_max_	[M-H]^−^	Main Snippet	Attempted Identification	Quantification		
		(nm)		ESI-MSn		(mg/g Extract)		
				[Intensity (%)]		EMP	EMNP	*p*-Value
1	6.05	324	353	191 (20), 179 (50), 173 (100), 135 (5)	4-*O*-caffeoylquinic acid ^1^	1.02 ± 0.05	0.40 ± 0.01	<0.01
2	6.92	325	353	191 (100), 179 (12), 173 (59), 135 (6)	5-*O*-caffeoylquinic acid ^1^	2.60 ± 0.16	1.18 ± 0.05	<0.01
3	10.41	325	367	191 (20), 173 (100), 135 (5)	4-*O*-*p*-coumaroylquinic acid ^2^	1.06 ± 0.03	1.92 ± 0.04	0.239
4	17.35	334	609	301 (100)	Quercetin-3-*O*-rutinoside ^3^	0.32 ± 0.01	0.656 ± 0.004	<0.01
5	18.57	326	515	353 (10), 191 (45), 179 (62), 173 (100), 135 (5)	3,4-*O*-dicaffeoylquinic acid ^1^	0.576 ± 0.001	0.24 ± 0.01	<0.01
6	19.94	326	515	353 (12), 191 (100), 179 (53), 173 (12), 135 (5)	3,5-*O*-dicaffeoylquinic acid ^1^	2.12 ± 0.03	1.97 ± 0.14	<0.01
7	22.47	326	515	353 (10), 191 (24), 179 (60), 173 (100), 135 (5)	4,5-*O*-dicaffeoylquinic acid ^1^	1.61 ± 0.02	0.38 ± 0.01	<0.01
Total phenolic compounds						9.31 ± 0.19	6.75 ± 0.24	0.599
Total phenolic acids						8.99 ± 0.18	6.09 ± 0.23	0.492
Total flavonoids						0.32 ± 0.01	0.656 ± 0.004	<0.01

EMP—yerba mate bio-residues; EMNP—yerba mate lyophilized leaves. Calibration curves used: ^1^—chlorogenic acid (168823x − 161172); ^2^—*p*-cumaric acid (301950x + 6966.7); ^3^—quercetin-3-*O*-rutinoside (y = 13343x + 76751). Statistical differences in means were obtained using the *t*-student test. *p*-value < 0.05 means significant differences between the concentrations of compounds.

**Table 2 foods-12-03241-t002:** Antibacterial (MIC and MBC mg/mL) and antifungal activity (MIC and MFC mg/mL) of extracts obtained from yerba mate samples.

**Antibacterial Activity**								
Extracts		** *S.a.* **	** *B.c.* **		** *L.m.* **	** *E.c.* **	** *S.t.* **	** *En.cl.* **
EMP	MIC	1	0.5		2	0.25	2	2
	MBC	2	2		2	0.5	2	4
EMNP	MIC	1	1		1	0.25	1	1
	MBC	2	2		2	0.5	2	2
Ampicillin	MIC	0.012	0.25		0.40	0.40	0.75	0.006
	MBC	0.025	0.40		0.50	0.50	1.20	0.012
**Antifungal Activity**								
Extracts		** *A.o.* **	** *A.n.* **	** *A.v.* **	** *P.f.* **	** *P.v.c.* **	** *P.o.* **	
EMP	MIC	0.5	0.5	0.25	0.5	0.5	0.25	
	MFC	1	1	0.5	1	1	0.25	
EMNP	MIC	0.5	0.5	0.25	0.5	0.5	0.125	
	MFC	1	1	0.5	1	1	0.125	
Ketoconazole	MIC	0.20	0.20	0.20	0.20	0.20	0.20	
	MFC	0.50	0.50	0.50	0.50	0.30	0.50	

EMP—yerba mate bio-residues; EMNP—yerba mate lyophilized leaves. *S.a.*: *Staphylococcus aureus*; *B.c.*: *Bacillus cereus*; *L.m.*: *Listeria monocytogenes*; *E.c.*: *Escherichia coli*; *En.cl.*: *Enterobacter cloacae*; *S.t.*: *Salmonella Typhimurium*; *A.o.*: *Aspergillus ochraceus*; *A.v.*: *Aspergillus versicolor*; *A.n.*: *Aspergillus niger*; *P.f.*: *Penicillium funiculosum*; *P.o.*: *Penicillium ochrochloron*; *P.v.c.*: *Penicillium verrucosum var. cyclopium* (food isolate). MIC: minimum inhibitory concentration; MBC: minimum bactericidal concentration; MFC: minimum fungicidal concentration.

**Table 3 foods-12-03241-t003:** Cytotoxic activity of extracts obtained from yerba mate samples (mean ± SD).

**Cytotoxic Activity**			
GI_50_ values (µg/mL)	EMP	EMNP	*p*-Value
HeLa	>400	>400	-
NCI-H460	>400	>400	-
MCF-7	196 ± 3	159.5 ± 0.1	<0.01
CaCo	162 ± 2	85.3 ± 0.2	<0.01
AGS	201 ± 7	238 ± 5	<0.01
VERO	>400	>400	-
**Anti-Inflammatory Activity**			
RAW 264.7	302 ± 10	258 ± 8	<0.01

EMP—yerba mate bio-residues; EMNP—yerba mate lyophilized leaves. HeLa: cervical carcinoma; NCI-H460: lung carcinoma; MCF7: breast carcinoma; CaCo-2: colorectal adenocarcinoma; AGS: gastric adenocarcinoma; VERO: non-tumoral line. Statistical differences in means were obtained using the *t*-student test. *p*-value < 0.05 means significant differences in the different extracts studied.

**Table 4 foods-12-03241-t004:** Antioxidant activity of extracts obtained from yerba mate samples (mean ± SD).

	Concentration		
Antioxidant Activity	EMP	EMNP	*p*-Value
TBARS	6.0 ± 0.3	5.8 ± 0.2	0.679
(EC_50_ values, µg/mL)			
OxHLIA _(Δt = 60 min)_	17.2 ± 0.8	23 ± 1	<0.01
(IC_50_ values, µg/mL)			
OxHLIA _(Δt = 120 min)_	41 ± 2	66 ± 2	<0.01
(IC_50_ values, µg/mL)			

EMP—yerba mate bio-residues; EMNP—yerba mate lyophilized leaves. EC_50_ values: Extract concentration corresponding to 50% of the antioxidant activity. Trolox (positive control) EC_50_ = 23 µg/mL (TBARS inhibition), 19.6 µg/mL (OXHLIA; Δt = 60 min) and 41 µg/mL (OXHLIA; Δt = 120 min). Statistical differences in means were obtained using the *t*-student test. *p*-value < 0.05 means significant differences between the different types of extract studied.

**Table 5 foods-12-03241-t005:** Color parameters measured in control pancakes (AC), with the addition of the extract rich in phenolic compounds (AEO) and with the addition of potassium sorbate (ASP), at times 0 and 3 days (mean ± SD).

	Time 0 Days			Time 3 Days			∆E	
	AC	AEO	ASP	AC	AEO	ASP		
L*	76.7 ± 1.1 ^a^	69.0 ± 2.3 ^c^	72.0 ± 2.3 ^b^	72.4 ± 1.6 ^a^	69.5 ± 0.9 ^b^	71.7 ± 2.1 ^a^	AC	5.4
a*	2.4 ± 0.1 ^b^	4.9 ± 0.2 ^a^	4.9 ± 0.2 ^a^	2.0 ± 0.1 ^b^	4.6 ± 0.2 ^a^	4.8 ± 0.2 ^a^	AEO	4.2
b*	32.0 ± 1.1 ^a^	32.3 ± 0.4 ^a^	32.3 ± 0.5 ^a^	28.8 ± 1.4 ^a^	28.1 ± 0.5 ^a^	28.1 ± 1.1 ^a^	ASP	4.2

L*—Luminosity; a*—chromatic axis from green (−) to red (+); b*—chromatic axis from blue (−) to yellow (+). AC—pancakes control without extract; AEO—pancakes enriched with the obtained extract rich in preservative phenolic compounds; ASP—pancakes with the addition of potassium sorbate. ∆E—total color difference. The results are expressed as mean ± standard deviation. The statistical differences in the means at times 0 and 3 days were obtained using one-way analysis of variance (ANOVA). In each row, different letters signify significant differences between the total amounts of compounds (*p* < 0.05). The statistical difference of the sample means with the addition of the extract rich in phenolic compounds was obtained by applying the *t*-student test, where *p*-value < 0.05 means a statistically significant difference.

**Table 6 foods-12-03241-t006:** Nutritional and chemical composition of control pancakes (CA), with the addition of the extract rich in phenolic compounds (AEO) and with the addition of potassium sorbate (ASP), at times 0 and 3 days, (mean ± SD and mean ± DP).

	Time 0 Days			Time 3 Days			∆AEO	*p*-Value
	AC	AEO	ASP	AC	AEO	ASP		
Humidity (g/100 g fw)	51.1 ± 0.1 ^a^	51.9 ± 0.1 ^a^	50.7 ± 0.1 ^a^	53 ± 0.1 ^a^	51.2 ± 0.1 ^b^	51.8 ± 0.1 ^b^	0.7	0.312
Proteins (g/100 g fw)	7.1 ± 0.2 ^a^	7.14 ± 0.01 ^a^	7.2 ± 0.1 ^a^	7.2 ± 0.3 ^a^	7.23 ± 0.01 ^a^	7.13 ± 0.03 ^a^	0.09	0.158
Ashes (g/100 g fw)	2.5 ± 0.1 ^a^	2.5 ± 0.1 ^a^	2.490 ± 0.003 ^a^	2.48 ± 0.05 ^a^	2.49 ± 0.04 ^a^	2.5 ± 0.1 ^a^	0.01	0.978
Fat (g/100 g fw)	3.17 ± 0.04 ^a^	3.2 ± 0.1 ^a^	3.2 ± 0.2 ^a^	3.2 ± 0.1 ^a^	3.230 ± 0.003 ^a^	3.2 ± 0.1 ^a^	0.03	0.323
Carbohydrates (g/100 g fw)	36.1 ± 0.2 ^a^	35.35 ± 0.03 ^a^	36.4 ± 0.1 ^a^	34.2 ± 0.3 ^a^	35.82 ± 0.03 ^a^	35.45 ± 0.03 ^a^	0.47	0.523
Energy (Kcal/100 g)	201.5 ± 0.5 ^a^	198.4 ± 1.1 ^a^	203.1 ± 1.1 ^a^	194.1 ± 0.8 ^a^	201.3 ± 0.2 ^a^	199.0 ± 0.3 ^a^	2.9	0.065
Energy (Kj/100 g)	844.4 ± 2.2 ^a^	831.3 ± 4.4 ^a^	851.1 ± 4.7 ^a^	813.3 ± 3.4 ^a^	843.4 ± 0.8 ^a^	833.9 ± 1.1 ^a^	12.1	0.065
**Sugars** (g/100 g fw)								
Sucrose	21.3 ± 0.9 ^a^	25.0 ± 0.5 ^c^	22.8 ± 0.9 ^b^	20.1 ± 0.7 ^a^	24.0 ± 0.6 ^c^	22.5 ± 0.2 ^b^	1.0	0.061
Total sugars	21.3 ± 0.9 ^a^	25.0 ± 0.5 ^c^	22.8 ± 0.9 ^b^	20.1 ± 0.7 ^a^	24.0 ± 0.6 ^c^	22.5 ± 0.2 ^b^	1.0	0.061
**Fat acids** (relative %)								
C16:0	26.1 ± 0.3 ^a^	26.9 ± 0.1 ^a^	26.5 ± 0.4 ^a^	26.2 ± 0.4 ^b^	27.2 ± 1.0 ^a^	26.9 ± 0.8 ^ab^	0.3	0.682
C18:0	7.4 ± 0.2 ^b^	8.1 ± 0.3 ^a^	8.2 ± 0.3 ^a^	7.2 ± 0.2 ^b^	8.1 ± 0.1 ^a^	8.1 ± 0.4 ^a^	0.0	0.964
C18:1n9	34.4 ± 0.3 ^b^	35.7 ± 0.1 ^a^	34.6 ± 0.5 ^b^	34.5 ± 0.7 ^b^	35.2 ± 0.6 ^a^	35.4 ± 0.6 ^a^	0.5	0.932
C18:2n6	19.7 ± 0.1 ^a^	19.1 ± 0.4 ^a^	20.6 ± 0.3 ^a^	19.4 ± 0.3 ^a^	18.9 ± 0.3 ^a^	18.9 ± 0.3 ^a^	0.2	0.734
SFA	41.8 ± 0.5 ^a^	42.6 ± 0.5 ^a^	42.5 ± 0.7 ^a^	41.7 ± 0.4 ^b^	43.4 ± 1.0 ^a^	43.0 ± 0.4 ^a^	0.8	0.695
MUFA	36.8 ± 0.3 ^b^	38.1 ± 0.1 ^a^	36.7 ± 0.4 ^b^	37.0 ± 0.7 ^a^	37.5 ± 0.7 ^a^	37.7 ± 0.6 ª	0.6	0.671
PUFA	21.4 ± 0.1 ^a^	19.3 ± 0.4 ^c^	20.7 ± 0.3 ^b^	21.3 ± 0.3 ^a^	19.1 ± 0.3 ^b^	19.3 ± 0.3 ^b^	0.2	0.889

AC—pancakes control without extract; AEO—pancakes enriched with the extract rich in preservative compounds; ASP—pancakes with the addition of potassium sorbate; Fw: fresh weight; Palmitic acid (C16: 0); Stearic acid (C18: 0); Oleic acid (C18: 1n9); Linoleic acid (C18: 2n6); SFA: saturated fatty acids; MUFA: monosaturated fatty acids; PUFA: polyunsaturated fatty acids. The results are expressed as mean ± standard deviation. The statistical differences in the means at times 0 and 3 days were obtained using one-way analysis of variance (ANOVA). In each row, different letters signify significant differences between the total amounts of compounds (*p* < 0.05). The statistical difference of the sample means with the addition of the extract rich in phenolic compounds was obtained by applying the *t*-student test, where *p*-value < 0.05 means a statistically significant difference.

## Data Availability

Data is contained within the article.

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
