# Peer review of "Use of Bio-Waste of Ilex paraguariensis A. St. Hil. (Yerba mate) to Obtain an Extract Rich in Phenolic Compounds with Preservative Potential"

_foods, 2023, doi:10.3390/foods12173241_

Round 1
Reviewer 1 Report
The manuscript presented a research of yerba-mate waste to obtain an extract rich in phenolic compounds with preservative potential in pancakes.
The introduction section should be more detailed. For example, "some/few studies" should be replaced with specific information from those studies.
The whole manuscript should be re-written to be presented in a presented in a well-structured manner. I suggest to merge results and discussion section in one section where discussion will follow the results straightly forward after the table and compare it with other research. It is hard to follow the discussion in the current form.
The references could be updated with recent publications (last 5 years).
The manuscript should be reviewed for editing of English language.
Author Response
Eliana Pereira
Centro de Investigação de Montanha
Campus Santa Apolónia
5300-253 Bragança - Portugal
Tel.: +351 273 303 382
e-mail: eliana@ipb.pt
Bragança, August 6th 2023
Review 1
The manuscript presented a research of yerba-mate waste to obtain an extract rich in phenolic compounds with preservative potential in pancakes.
The introduction section should be more detailed. For example, "some/few studies" should be replaced with specific information from those studies.
A: Thank you for your suggestion. The change in introduction section was done (line 64-75).
The whole manuscript should be re-written to be presented in a presented in a well-structured manner. I suggest to merge results and discussion section in one section where discussion will follow the results straightly forward after the table and compare it with other research. It is hard to follow the discussion in the current form.
A: Thank you for your consideration, the intro has been changed (line 64-75), but the structure of the manuscript not, because its adheres to the journal's guidelines, where it divides into sections for the obtained results and subsequent discussion.
The references could be updated with recent publications (last 5 years).
A: Thank you for your consideration. The possible references have been updated, and the remaining ones are highly relevant to support the crucial information that was not found in current references.
The manuscript should be reviewed for editing of English language.
A: All the manuscript was revised.

Reviewer 2 Report
These paper’s findings are encouraging and they evidence the use of bio-waste as a source of bioactive compounds for development of promising functional ingredients and environmentally sustainable food products. Overall, I think the paper is very interesting and need major revision. The manuscript is well structured and well written. The suggestions are described below.
1) Line 22-23: “.It was verified the absence of toxicity in both extracts”. Please, rewrite. The authors found extracts cytotoxicity in MCF-7, CaCo and AGS cancer cells.
2) Line 72-73: Please standardize: bio-waste or biowaste?
3) Why did authors choose an 80:20 ratio of ethanol and water for extraction?
4) Please detail the extract concentrations tested in the antimicrobial, cytotoxic, anti-hemolytic and anti-inflammatory activity analyses.
5) Please organize the layout of tables 5 and 6 so that the words and values are aligned.
6) I suggest the authors carry out a statistical analysis of the colorimetric parameters between the same sample of pancakes before and after the storage period, to verify if there was a difference (as the same way as they did in table 6 - comparison between the same sample, before and after storage).
7) In table 6, I suggest that instead of repeating the value of AEO 0 days and AEO 3 days, present the delta value, followed by the p-value.
8) Line 327-342: This information is more like a result than a discussion.
9) The discussion about chemical composition is dense, plastered and tiring. Focus on the main compounds found and relate them to the biological properties found in the work itself. For example, which compounds present in extracts may be associated with antimicrobial, antiproliferative activity?
10) Line 478: Please, put the GI50 values in parentheses for MCF-7, CaCo and AGS cells.
11) Line 571-579: Avoid very short paragraphs. You can join them.
Author Response
Eliana Pereira
Centro de Investigação de Montanha
Campus Santa Apolónia
5300-253 Bragança - Portugal
Tel.: +351 273 303 382
e-mail: eliana@ipb.pt
Bragança, August 6th 2023
Manuscript ID: foods-2474650
Title: Use of bio-waste of Ilex paraguariensis A. St.Hil. (yerba-mate) to obtain an extract rich in phenolic compounds with preservative potential
Review 2
These paper’s findings are encouraging and they evidence the use of bio-waste as a source of bioactive compounds for development of promising functional ingredients and environmentally sustainable food products. Overall, I think the paper is very interesting and need major revision. The manuscript is well structured and well written. The suggestions are described below.
A: We are thankful for taking the time to thoroughly read the manuscript and for providing us with your valuable comments and suggestions.
1) Line 22-23: “.It was verified the absence of toxicity in both extracts”. Please, rewrite. The authors found extracts cytotoxicity in MCF-7, CaCo and AGS cancer cells.
A: Thank you for your comment, the sentence has been rewritten (line 22-23). However, when we talk about cytotoxicity, tumor cell lines and non-tumor cell lines are considered; but when talking about toxicity we are only talking about non-tumor cell lines or primary non-tumor cell culture.
2) Line 72-73: Please standardize: bio-waste or biowaste?
A: Thank you for your suggestion, the term "bio-waste" has been standardized.
3) Why did authors choose an 80:20 ratio of ethanol and water for extraction?
A: Thank you for your question. The 80:20 (v/v) ethanol:water ratio was selected based on previous research conducted by our team, which demonstrated that this proportion yielded the best results in phenolic compound extraction. Ethanol was chosen as the extraction solvent due to its non-toxic nature, as the extracted compounds will be incorporated into food products.
4) Please detail the extract concentrations tested in the antimicrobial, cytotoxic, anti-hemolytic and anti-inflammatory activity analyses.
A: Thank you for tour comment. However, in section 2.3 all this information is available.
5) Please organize the layout of tables 5 and 6 so that the words and values are aligned.
A: Thank you for your comment, the tables have been corrected.
6) I suggest the authors carry out a statistical analysis of the colorimetric parameters between the same sample of pancakes before and after the storage period, to verify if there was a difference (as the same way as they did in table 6 - comparison between the same sample, before and after storage).
A: The statistical treatment for the color parameters is in the table and it was done in the same way as table 6. The statistical differences in the means at times 0 and 3 days were obtained through one-way analysis of variance (ANO-VA). In each row, different letters signify significant differences between the total amounts of compounds (p < 0.05). The statistical difference of the sample means with the addition of the extract rich in phenolic compounds was obtained by applying the t-student test, where p-value < 0.05 means a statistically significant difference.
7) In table 6, I suggest that instead of repeating the value of AEO 0 days and AEO 3 days, present the delta value, followed by the p-value.
A: Thank you for your suggestion, the table has been adjusted.
8) Line 327-342: This information is more like a result than a discussion.
A: Thank you for your comment, the paragraph has been removed.
9) The discussion about chemical composition is dense, plastered and tiring. Focus on the main compounds found and relate them to the biological properties found in the work itself. For example, which compounds present in extracts may be associated with antimicrobial, antiproliferative activity?
A: Thank you for your comment. This section was restructured.
10) Line 478: Please, put the GI50 values in parentheses for MCF-7, CaCo and AGS cells.
A: Thank you for your comment, the information has been added throughout the paragraph. However all the results and discussion sections were restructured and this information was change to the results section.
11) Line 571-579: Avoid very short paragraphs. You can join them.
A: Thank you for your comment, the paragraphs have been joined.

Reviewer 3 Report
Why was it decided to use the extraction conditions and not the evaluation of different methods and mix of conditions?
Will the E. coli tests be re-evaluated? As it is a pathogen of importance in food?
Why was it decided to evaluate against fungi? Are they the ones attacking the final incorporation proposal?
What happens to the compounds during cooking?
Author Response
Review 3
- Why was it decided to use the extraction conditions and not the evaluation of different methods and mix of conditions?
A: In this study, sampling was not mixed because the leaves were used only as a control. We chose maceration as the extraction technique because it is economically more feasible for industrial purposes. Additionally, we aim to conduct a study that closely resembles a viable pilot application.
- Will the E. coli tests be re-evaluated? As it is a pathogen of importance in food?
A: The strain E. coli is a common bacterium found in water. If the water is used both for irrigation and for later cleaning, before the drying process. So, it is a strain of great nutritional importance.
- Why was it decided to evaluate against fungi? Are they the ones attacking the final incorporation proposal?
A: If the objective is to develop an extract with preservative potential, its bactericidal, bacteriostatic, fungicidal and fungistatic capacity must be tested. Since the growth of these pathogens in food are one of the main causes of its degradation.
- What happens to the compounds during cooking?
A: If this question is directed to phenolic compounds, this parameter (influence of temperature on phenolic compounds) has not yet been studied. The stability study (which includes pH, temperature and light studies) will be done later.

Reviewer 4 Report
Dear Authors,
In the study, you compared extracts of dried yerba-mate (Ilex paraguariensis) with biowaste of yerba-mate leaves and assessed the possibility of using them as a preservative/functional ingredient for a pastry product (pancakes). You determined the profile of phenolic compounds and evaluated the antioxidant, anti-inflammatory, antimicrobial and cytotoxic activity. Although you have carried out a wide range of analyses, the peer-reviewed manuscript contains numerous language errors and needs improvement in the description of the results and discussion.
The errors that need to be corrected are listed below:
-
In the introduction, the properties of yerba mate, proven in scientific articles, should be mentioned in more detail.
-
The English language needs to be corrected as the manuscript contains numerous grammatical errors, unnecessary commas and incorrectly formulated sentences
-
Why are the concentrations of the extracts for the antibacterial (10 mg/mL), cytotoxic (8 mg/mL) and antioxidant (5 mg/mL) tests different? What is the reason for using different concentrations? I think there are also errors regarding the solvent used in particular types of analyses.
-
Why were the extracts dissolved in water and not in culture medium for cytotoxicity analyses?
-
No information about the source of obtaining/purchasing cell lines - please add this information in the methodology section
-
The sentence on lines 151-152 should be corrected as it is not understandable.
-
Line 152: "Cell cultures were made..." is not correct
-
Lines 163 and 168: the word "translate" is not used correctly - please replace it with another word
-
“ex vivo”, “in vivo” and “in vitro” should be italicized throughout the manuscript
-
Lines 163-164: the wording "responsible for 50% of antioxidant activity" should be corrected as it is incomprehensible in this form
-
Subitem names e.g.< 2.4 Incorporation of the extract rich in phenolic compounds, obtained from the bio-residues 170 of yerba-mate, in a pastry product – pancakes “ are too long - it should be corrected
-
Lines 248-249 - no reference to literature items.
-
The description of the methodology states that the tested samples subjected to antibacterial analysis were dissolved in the culture medium, and the description of the results indicated hydroethanolic extracts - it should be clarified
-
Table descriptions should be corrected as they are illegible
-
The description of the antibacterial test methodology should indicate what concentrations of the tested samples were used during the tests
-
The "Results" section lacks at least a brief description of the results obtained - they are shown only in tables
-
Line 326: the word "namely" was used incorrectly
-
Line 435. The full name of the abbreviation En.cl. should be given as for other bacterial strains
-
The authors should enrich the discussion with an indication of potential antimicrobial mechanisms of the tested samples
-
The statement "In this sense" appearing in the text of the manuscript seems to be not entirely correct - it suggests replacing it with another statement
-
Possible mechanisms of inhibition of tumor cell proliferation should be mentioned in the discussion
-
The discussion is quite poorly written - references to the works of other authors are too extensively described, without indicating the possible mechanisms of activity of the tested extracts
-
The discussion contains too many short paragraphs - reduce the number of paragraphs and write in the form of compact text. Authors can use paragraphs that each refer to different properties.
-
There is no indication in the discussion of biologically active compounds that may be responsible for the differences in the antioxidant properties of the tested samples
-
It is not necessary to use both the species name and the abbreviation of the bacterial strain (e.g. Escherichia coli (E.c.)) in different parts of the manuscript - it is enough only in the first place where the abbreviation appears for the first time - this should be corrected
-
In the "Conclusion" part, the number of paragraphs should be reduced!
I am not qualified to judge the quality of the English in this article, but I think it needs extensive English language editing
Author Response
Review 4
Dear Authors,
In the study, you compared extracts of dried yerba-mate (Ilex paraguariensis) with biowaste of yerba-mate leaves and assessed the possibility of using them as a preservative/functional ingredient for a pastry product (pancakes). You determined the profile of phenolic compounds and evaluated the antioxidant, anti-inflammatory, antimicrobial and cytotoxic activity. Although you have carried out a wide range of analyses, the peer-reviewed manuscript contains numerous language errors and needs improvement in the description of the results and discussion.
The errors that need to be corrected are listed below:
- In the introduction, the properties of yerba mate, proven in scientific articles, should be mentioned in more detail.
A: Thank you for your suggestion, this information has been added.
- The English language needs to be corrected as the manuscript contains numerous grammatical errors, unnecessary commas and incorrectly formulated sentences
A: Thank you for your consideration, the manuscript has been revised.
- Why are the concentrations of the extracts for the antibacterial (10 mg/mL), cytotoxic (8 mg/mL) and antioxidant (5 mg/mL) tests different? What is the reason for using different concentrations? I think there are also errors regarding the solvent used in particular types of analyses.
A: The evaluation of bioactivities involves several different assays, testing various biological actions. These methods are not comparable in any aspect, such as the solvents used, culture media, initial test concentrations, etc. The application of these methodologies in our research group underwent a rigorous optimization process, tested on different matrices, to fully validate the entire procedure. Furthermore, these methodologies have been adopted and replicated by other research groups, further attesting to their reliability and rigor. Therefore, there are no errors in the solvents used, as their selection depends on the compounds being analyzed. Different concentrations are characteristic of each method, adjusted to each specific matrix.
- Why were the extracts dissolved in water and not in culture medium for cytotoxicity analyses?
A: Firstly, the extracts used in this study are water-soluble, making them easily dissolved in water. On the other hand, for this cytotoxicity assay, different culture media are employed, tailored to specific cell lines. Using one of these culture media to dissolve the extracts would restrict their applicability.
- No information about the source of obtaining/purchasing cell lines - please add this information in the methodology section
A: Thank you for the comment, the information has been added.
- The sentence on lines 151-152 should be corrected as it is not understandable.
A: Thank you for your comment, the sentence has been corrected (line 161-162).
- Line 152: "Cell cultures were made..." is not correct
A: Thank you for your comment, the sentence has been corrected (line 163).
- Lines 163 and 168: the word "translate" is not used correctly - please replace it with another word
A: Thank you for your suggestion, the sentence has been rewritten (line 173 and 178).
- “ex vivo”, “in vivo” and “in vitro” should be italicized throughout the manuscript
A: Thank you for your comment, the words have been corrected.
- Lines 163-164: the wording "responsible for 50% of antioxidant activity" should be corrected as it is incomprehensible in this form
A: Thank you for your suggestion, the sentence has been rewritten (line 173-174).
- Subitem names e.g.< 2.4 Incorporation of the extract rich in phenolic compounds, obtained from the bio-residues 170 of yerba-mate, in a pastry product – pancakes “are too long - it should be corrected
A: Thank you for your comment, the sub-item name has been replaced.
- Lines 248-249 - no reference to literature items.
A: Thank you for the comment, the information has been added (line 290-291).
- The description of the methodology states that the tested samples subjected to antibacterial analysis were dissolved in the culture medium, and the description of the results indicated hydroethanolic extracts - it should be clarified
A: Thank you for your comment. The description was clarified.
- Table descriptions should be corrected as they are illegible
A: Thank you for your comment. The descriptions were changed.
- The description of the antibacterial test methodology should indicate what concentrations of the tested samples were used during the tests
A: Thank you for the comment, the information has been added (line 292)
- The "Results" section lacks at least a brief description of the results obtained - they are shown only in tables
A: Thank you for the comment, this has been resolved.
- Line 326: the word "namely" was used incorrectly
A: Thank you for your suggestion, the word has been rewritten.
- Line 435. The full name of the abbreviation En.cl. should be given as for other bacterial strains
A: Thank you for your comment, it has been corrected.
- The authors should enrich the discussion with an indication of potential antimicrobial mechanisms of the tested samples
A: Thanks for your comment. All the discussion section was restructured.
- The statement "In this sense" appearing in the text of the manuscript seems to be not entirely correct - it suggests replacing it with another statement
A: Thank you for your comment. The changes were made in the document.
- Possible mechanisms of inhibition of tumor cell proliferation should be mentioned in the discussion
A: Thank you for the comments, the information has been added in the discussion section.
- The discussion is quite poorly written - references to the works of other authors are too extensively described, without indicating the possible mechanisms of activity of the tested extracts
A: Thank you for the comments. This section was restructured.
- The discussion contains too many short paragraphs - reduce the number of paragraphs and write in the form of compact text. Authors can use paragraphs that each refer to different properties.
A: Thank you for your comment, the paragraphs have been joined and compressed.
- There is no indication in the discussion of biologically active compounds that may be responsible for the differences in the antioxidant properties of the tested samples
A: Thank you for your evaluation. The information was added in discussion section (line 603).
- It is not necessary to use both the species name and the abbreviation of the bacterial strain (e.g. Escherichia coli (E.c.)) in different parts of the manuscript - it is enough only in the first place where the abbreviation appears for the first time - this should be corrected
A: Thank you for your comment, it was corrected.
- In the "Conclusion" part, the number of paragraphs should be reduced!
A: Thank you for your comment, the paragraphs have been joined and compressed.
